



# Formation kinetics and mechanism of ozone and secondary organic aerosols from photochemical oxidation of different aromatic hydrocarbons: dependence of NOx and organic substituent

Hao Luo[1,2,★], Jiangyao Chen[1,2,★], Guiying Li[1,2], and Taicheng An[1,2]

[1]Guangdong Key Laboratory of Environmental Catalysis and Health Risk Control, Guangdong-Hong Kong-Macao Joint Laboratory for Contaminants Exposure and Health, Institute of Environmental Health and Pollution control, Guangdong University of Technology, Guangzhou 510006, China
[2]Guangzhou Key Laboratory of Environmental Catalysis and Pollution Control, Key Laboratory of City Cluster Environmental Safety and Green Development, School of Environmental Science and Engineering, Guangdong University of Technology, Guangzhou 510006, China

★ These authors contributed equally to this work.

*Correspondence to*: Taiceng An (antc99@gdut.edu.cn)

**Abstract.** Aromatic hydrocarbons (AHs) contribute significantly to ozone and secondary organic aerosol (SOA) formation in atmosphere, but formation mechanisms are still unclear. Herein, photochemical oxidation of nine AHs was investigated in chamber. Only small amount of ozone was produced from direct photochemical oxidation of AHs, while fewer AH substituent number resulted in higher concentrated ozone. Addition of NOx increased ozone and SOA production. Synergetic effect of accelerated NO2 conversion and NO reaction with AHs boosted ozone and volatile intermediate formation. Promoting AH concentration in VOC/NOx ratio further increased formation rates and concentrations of both ozone and SOA. Additionally, ozone formation was enhanced with increasing AH's substituent number but negligibly affected by their substituent position. Differently, SOA yield decreased with increased substituent number of AHs, but increased with ortho methyl group substituted AHs. Model fitting and intermediate consistently confirmed that increasing substituent number on phenyl ring inhibited generating dicarbonyl intermediates, which however were preferentially produced from oxidation of ortho methyl group substituted AHs, resulting in different changing trend of SOA yield. The restrained oligomerization by increased substituent number was another main cause for decreased SOA yield. These results are helpful to understand photochemical transformation of AHs to secondary pollutants in real atmosphere.

## 1 Introduction

As an abundant group of volatile organic compounds (VOCs), aromatic hydrocarbons (AHs) are important precursors of ozone ($O_3$) and secondary organic aerosols (SOA) in atmospheric environment (Peng et al., 2017;Tong et al., 2020), directly or indirectly threatening air quality and public health (Henze et al., 2008;Lane et al., 2008;Yang et al., 2016). Atmospheric AHs mainly come from anthropogenic sources, such as industrial emission and motor vehicle emissions (Luo et al.,



2020a;Sun et al., 2018;Chen et al., 2020;An et al., 2014;He et al., 2015), and these emitted AHs are commonly composed of single phenyl ring with less than four methyl groups (e.g., toluene, xylene) or ethyl groups (e.g., ethylbenzene) (Han et al., 2019;Hu et al., 2015;Chen et al., 2019). It is also found that photochemical oxidation of these AHs are sensitive to reaction conditions (e.g., VOC/NO$_x$ ratio (Odum et al., 1996;Metzger et al., 2008;Bloss et al., 2005a;Carter and Heo, 2013)), ultimately influencing the formation kinetics and mechanism of O$_3$ and SOA from AH oxidation (Borrás and Tortajada-Genaro, 2012;Sato et al., 2010;Cocker III et al., 2001;Ji et al., 2017;Jia and Xu, 2018).

Previous experimental simulation studies have confirmed that relatively high VOC/NO$_x$ ratio had an inhibitory effect on SOA productivity (Wang et al., 2015), while VOC/NO$_x$ ratio might also influence O$_3$ formation (Wang et al., 2016). From field observation, the actual ratio of VOC/NO$_x$ in the atmosphere is always changed with the variation of the seasons (Geng et al., 2008;Zou et al., 2015;Li et al., 2013;Seinfeld, 1989). Nevertheless, although it is of important environmental significance, the studies about influence of VOC/NO$_x$ ratio on photochemical oxidation of AHs to form O$_3$ and SOA are mostly focused on the varied NO$_x$ concentration. The effects of AH content as well as substitute groups of AHs to the formation kinetics and mechanism of O$_3$ and SOA are still unconcerned. Therefore, in view of the complexity of the real atmosphere, it is very necessary to effectively simulate atmospheric photochemical reactions at different VOC/NO$_x$ ratios in laboratory smog chamber and explore the formation kinetics and mechanism of O$_3$ and SOA from AHs at different concentrations.

Usually, ·OH-initiated reactions have been confirmed to dominate in AH photochemical oxidation (Ji et al., 2018), in which the reaction rate constant increases with increased substituent number of AHs (Atkinson and Arey, 2003;Aschmann et al., 2013;Glasson and Tuesday, 1970). The important role of substituent position has also been observed in the ·OH-initiated alkane and alkene oxidation (Atkinson, 2007;Ziemann, 2011). All these previous studies inspire us that the influence of substituent, including number and position, on the photochemical transformation of AHs to O$_3$ and SOA cannot be ignored. However, although both O$_3$ and SOA generated from different AHs have been studied in laboratory smog chamber simulations, the role of AH's substituent to the formation kinetics and mechanism of O$_3$ and SOA as well as their relationship with the oxidation intermediates have not been systematically investigated and established.

In this work, nine AHs with different substituent number and position (e.g., benzene, toluene, ethylbenzene, m-xylene, o-xylene, p-xylene, 123-trimethylbenzene, 124-trimethylbenzene, 135-trimethylbenzene) were chosen to study their photochemical oxidation behavior in an indoor smog chamber system, to compare the formation activity in O$_3$ and SOA. The influences of NO$_x$, AH concentration and AH substituent on the formation kinetics of O$_3$ and SOA were studied in detail. All volatile intermediates were qualitatively and quantitatively online analyzed to propose their potential contribution to the formation of O$_3$ and SOA. The relationship between AH structure, intermediate, and production of O$_3$ and SOA were established to reveal the transformation mechanism of AHs to O$_3$ and SOA. The results of this work will further elucidate the photochemical behavior of AHs in the atmosphere and provide reliable experimental data for modeling and prediction in the future.



## 2 Experimental

### 2.1. Photochemical oxidation experiment

All experiments were conducted in GDUT-DRC dual-reactor chamber with two 2 m³ pillow-shaped reactors and the detail description of the reactor was reported in our early work (Luo et al., 2020b). The experimental relative humidity and temperature was set as < 5% and around $303 \pm 1$ K, respectively. No inorganic seed aerosol was supplied in this work. A total of 60 black lamps (40 W, F40BL, GE, USA) were equipped to provide UV light source, and the light intensity in the dual-reactor were determined to be 0.161 min⁻¹ (left) and 0.169 min⁻¹ (right) by using the $NO_2$ photolysis rate constant (Luo et al., 2020b).

Nine AHs (benzene, 99.5%; toluene, 99.0%; ethylbenzene, 99.8%; m-xylene, 99.0%; o-xylene, 99.0%; p-xylene, 99.0%; 1,2,3-trimethylbenzene (123-TMB), 90.0%; 1,2,4-trimethylbenzene (124-TMB), 98.0%; 1,3,5-trimethylbenzene (135-TMB), 97.0%) purchased from Aladdin Industrial Co., Ltd. (USA) and certain amount of $NO_2$ were directly injected into the reactor to conduct the photochemical oxidation experiments. Before turning on the light, all AHs and $NO_2$ were injected with the background gas flow and adjusted to stable for one hour.

### 2.2. Organic gas measurement

The concentrations of AHs and their oxidation products were all measured online using proton-transfer reaction time-of-flight mass spectrometry (PTR-ToF-MS, Ionicon Analytik Inc., Austria). In the setting model, all gaseous organics with proton affinity greater than $H_2O$ including AHs, hydrocarbons, acids and carbonyl groups, were all measured quantitatively and qualitatively. By processing software TOF-DAQ (Tofwerk AG, Switzerland), it recorded the material with m/z≤240 and the original signal strength was converted into ppb concentration by formula (Lindinger et al., 1998). Before sampling and measurement, PTR-ToF-MS was calibrated and the instrument was calibrated once a week during all measurement period (Han et al., 2019). Detailed parameters of the instrument and the yield calculation formula of products were given in Supporting Information (SI).

### 2.3. Inorganic gas measurement

The real-time concentrations of NO, $NO_2$, and $NO_x$ were all spontaneously monitored with $NO_x$ analyzer (Model 42i, Thermo Scientific Inc., USA), and the real-time concentration of $O_3$ was monitored with $O_3$ analyzer (Model 49i, Thermo Scientific Inc., USA). All devices were calibrated weekly using gas calibrator (Model 146i, Thermo Scientific Inc., USA).

### 2.4. Particle measurement

Particle size distribution was measured by scanning mobility particle sizer spectrometer (SMPS, TSI Inc., USA) equipped with electrostatic classifiers (EC, Model 3082, TSI Inc., USA) and long differential mobility analyzer (DMA) (Model 3081, TSI Inc., USA), or an optional nano DMA (Model 3085, TSI Inc., USA) and a condensation particle counter (CPC, Model





3776, TSI Inc., USA). The velocities of sheath gas and aerosol flows were set at 3.0 and 0.3 L min$^{-1}$, respectively. Under this
    setting, the particle size range was observed from 13.8 to 723.4 nm. The yield calculation formula of SOA was given in SI.

## 3 Results and discussion

### 3.1 Formation kinetics and mechanism of O$_3$ and SOA without NO$_x$

The directly photochemical oxidation of nine AHs was first conducted to evaluate the formation potential of O$_3$ and SOA.

No SOA was detected within 480 min's reaction duration. The O$_3$ concentration increased steadily from 0 ppb to 16 for 135-
    TMB and 28 ppb for ethylbenzene within 420 min (Fig. 1a). The peak concentration of O$_3$ for these nine AHs followed the
    trend of (ethylbenzene, toluene, benzene, $23 - 28$ ppb) > (m-, o-, p-xylene, $18 - 21$ ppb) > (123-, 124-, 135-TMB, $16 - 20$
    ppb). Clearly, fewer number of AH substituent resulted in generation of higher concentrated O$_3$. This may be because that
    the substituent number of AHs determined their reactivity, while the reaction rate constants with ·OH could be applied to

evaluate the reactivity. Previous studies indicated that the rate constants of AHs with ·OH was found obeying the order of
    toluene $(5.61 \pm 0.08) \times 10^{-12}$ cm$^3$ molecule$^{-1}$ s$^{-1}$ < xylene $(7.4 - 14) \times 10^{-12}$ cm$^3$ molecule$^{-1}$ s$^{-1}$ < TMB $(14 - 31) \times 10^{-12}$ cm$^3$
    molecule$^{-1}$ s$^{-1}$ at $304 \pm 1$ K (Doyle et al., 1975;Anderson and Hites, 1996). Combining with our results herewith, it is solidly
    concluded that AHs with fewer substituent number showed lower reactivity, and then resulting in higher concentrated O$_3$
    formation.

Furthermore, to figure out the formation mechanism of O$_3$ from direct AH oxidation, the correspondingly volatile
    intermediates were also monitored. Toluene was taken as an example to illustrate the concentration variation of volatile
    intermediates. As Fig. 1b shows, with decrease of toluene's concentration from 996 to 944.5 ppb, the concentrations of nine
    intermediates increased at different degrees. The concentrations of m/z = 45 (m45, acetaldehyde), m/z = 47 (m47, formic
    acid) and m/z = 61 (m61, acetic acid or glycolaldehyde) increased faster than others, and peaked at $5 - 9$ ppb within 450 min,

indicating easily oxidation of toluene to small molecular carbonyl products. The peak concentration of m/z = 99 (m99, 3-
    methyl-2(5H)-furanone or 4-keto-2-pentenal) reached 4.2 ppb, while the productions of m/z = 31 (m31, formaldehyde), m/z
    = 59 (m59, glyoxal), m/z = 73 (m73, methylglyoxal) were at the same level (ca. 2.4 ppb). The concentrations of some
    intermediates including m/z = 85 (m85, butanedione), m/z = 87 (m87, butenedione) and m/z = 111 (m111, hexene diketone)
    were lower than 0.8 ppb within 450 min's reaction duration. Similar variation trends of volatile intermediates were observed

from other eight AHs (Fig. S1).

    It was worth mentioning that most of above intermediates were well-known precursors of O$_3$ and SOA (Li et al.,
    2016;Ji et al., 2017;Nishino et al., 2010). However, the formation of SOA was not observed in this study. Two reasons might
    be involved. Since this study was carried out at low RH (< 5%) and without seed particles, no SOA precursor oligomers
    existed. Furthermore, the concentrations of produced intermediates were too low to trigger the initial nucleation reaction and

then generate SOA under low RH condition. Therefore, SOA formation could not be observed in the NO$_x$-free
    photochemical oxidation of these nine AHs. In general, tropospheric O$_3$ was mainly from NO$_2$ photolysis and the existence





of AHs could enhance $O_3$ formation. However, when the absent of $NO_x$ in this study, the low concentrated $O_3$ was observed from AH photochemical oxidation. The possible contributors of these $O_3$ might be intermediates such as carbonyl compounds. In all, our results indicated that direct photochemical transformation of AHs to $O_3$ actually occurred and should

be taken into consideration in the atmospheric environment.

### 3.2. Formation kinetics and mechanism of $O_3$ in the presence of $NO_x$

To further explore the role of $NO_x$ in $O_3$ formation during photochemical oxidation of AHs, about $160 \pm 10$ ppb of $NO_2$ was added into the reactor. Under this condition, the generated $O_3$ was found significantly increasing and the $O_3$ peak concentrations ranged from 230 to 440 ppb within 100 to 250 min (Fig. 2). All these data were about $200 - 400$ ppb higher

than those obtained in the absence of $NO_x$ (Fig. 1a), indicating quick enhancement of $NO_x$ to $O_3$ formation. In this work, the added $NO_2$ was firstly photolyzed under 360 nm's light irradiation to from NO and $O(^3P)$ (Eq. 1). Then, the latter was oxidized to form $O_3$ (Eq. 2), which further reacted with NO to form $NO_2$ (Eq. 3). Meanwhile, AHs were photochemically oxidized to form $RO_2$ and $HO_2$, both of which then reacted with NO to form $NO_2$ (Eqs. 4 and 5). Clearly, the presence of AHs could compete with $O_3$ for the NO reaction and reduce the consumption of $O_3$. The synergetic effect of direct $NO_2$

conversion and AH's competition reaction led to the boosting formation of $O_3$ in the presence of both AHs and $NO_x$.

$$NO_2 + h\nu \ (\lambda = 360 \ nm) \rightarrow NO + O(^3P) \quad (1)$$

$$O(^3P) + O_2 \rightarrow O_3 \quad\quad\quad\quad (2)$$

$$NO + O_3 \rightarrow NO_2 \quad\quad\quad\quad (3)$$

$$RO_2 + NO \rightarrow NO_2 + RO \quad\quad (4)$$

$$HO_2 + NO \rightarrow NO_2 + {}^{\bullet}OH \quad\quad (5)$$

Furthermore, the effect of AH content on the $O_3$ formation in the presence of $NO_x$ (e.g., $VOC/NO_x$ ratio) was investigated. Here, the concentration of $NO_x$ was maintained constantly and that of AH was gradually increased. For toluene (Fig. 3a), $O_3$ peak concentration of 250 ppb was achieved after reaction for 420 min under $VOC/NO_x$ ratio of 2.47. When increasing $VOC/NO_x$ ratio to 6.29, the time needed achieving higher peak $O_3$ concentration of 280 ppb was shortened to be

150 min. All these data confirmed that $O_3$ formation rate and concentration were both accelerated with increased AH concentration. Similar results of shorter reaction time leading to higher $O_3$ concentration were observed for the rest AHs (Fig. S2). Increasing AH concentration would result in the enhanced formation of $RO_2$ and $HO_2$, both of which reacted with NO to save the $O_3$ consumption. Meanwhile, the photolysis of $NO_2$ to form NO and then $O_3$ was also accelerated. Both of these reasons were responsible for the fast-enhanced formation of $O_3$ with the increased AH concentration in $VOC/NO_x$ ratio.

To study the effect of AH's substituent on $O_3$ formation, the $O_3$ peak concentrations of nine AHs obtained at the same $VOC/NO_x$ ratio were compared. As Fig. 3b shows, the $O_3$ peak concentrations of nine AHs followed the order of TMB $(366.4 - 431.2 \ ppb) >$ xylene $(290.6 - 365.7 \ ppb) >$ toluene and ethylbenzene $(246.7 - 291.7 \ ppb) >$ benzene $(187.3 - 231.2$





ppb). Clearly, the $O_3$ peak value was positively correlated with the number of AH's substituent, suggesting AHs with more substituent possessed higher $O_3$ production potential at the same VOC/$NO_x$ ratio. In previous studies, $O_3$ concentrations from

AH oxidation with the presence of $NO_x$ were reported as follows: $210 - 320$ ppb for benzene, $160 - 300$ ppb for toluene, $550 - 700$ ppb for ethylbenzene, 400 ppb for m-xylene, $300 - 600$ ppb for o-xylene, $300 - 350$ ppb for p-xylene, $340 - 470$ ppb for 123-TMB, 500 ppb for 124-TMB and $400 - 700$ ppb for 135-TMB (Wang et al., 2016;Luo et al., 2019;Li et al., 2018;Xu et al., 2015;Jia and Xu, 2013;Lu et al., 2009;Carter, 1997). However, these studies only focused on one or several AHs, and the relationship between AH substituent and $O_3$ formation was still not understood. Our results of $O_3$ concentration were

comparable to those in the previous studies. Furthermore, the results obtained in this study clearly confirmed that increasing substituent number of AH correspondingly increased $O_3$ concentration. It was also noticed that the $O_3$ peak concentrations of xylene or TMB isomers were in the same range, suggesting negligible effect of substituent position of AHs to their $O_3$ formation.

### 3.3. Accelerated formation of SOA in the presence of $NO_x$

Besides $O_3$, the effect of AH concentration on formation kinetics of SOA with the presence of NOx was also investigated. As Fig. 4 shows, from photochemical oxidation of toluene, the peak number concentration of SOA increased from $2.0 \times 10^4$ to $5.5 \times 10^4$ particle cm$^{-3}$ with increase of VOC/$NO_x$ ratio from 2.37 to 5.58. The time achieving above concentration was shortened from 250 to 120 min, while the median particle size range also increased from $300 - 400$ to $400 - 500$ nm. Similar results of shorter time leading to higher concentration and larger particle size for SOA were observed for other eight AHs

with the increasing VOC/$NO_x$ ratio (Figs. S3-S10).

Previous studies reported the enhanced SOA yield by increased $NO_x$ concentration (Zhao et al., 2018;Hurley et al., 2001;Song et al., 2007;Sarrafzadeh et al., 2016). This was because that $NO_x$ mainly influenced the distribution of oxidation products by affecting the $RO_2$ reaction equilibrium, where $RO_2$ easily converted to low-volatile ROOH or ROOR and thus resulted in the nucleation of new particles (Sarrafzadeh et al., 2016). However, in this study, we kept the $NO_x$ concentration

unchanged, and modified the initial concentration of AHs. The increased AHs could lead to promoted $RO_2$ formation, resulting in more low-volatile products formation. The accumulation of low-volatile products promoted the nucleation of particulate matter and finally increased the yield of SOA.

The particle number and mass concentrations of SOA generated from nine AHs were further compared to evaluate the effect of AH's substituent on the SOA formation. With the increase of substituent number, the number concentration of SOA

decreases (e.g., from $6.9 \times 10^3$ particle m$^{-3}$ for 135-TMB to $7.8 \times 10^4$ particle m$^{-3}$ for toluene) (Fig. 5a). With the progress of the reaction, the mass concentration of SOA increased, and the increase of substituent number shortened the time achieving the peak mass concentration (Fig. 5b). These results revealed that the increase of substituent number of AHs increased SOA's mass concentration but decreased its particle number. AHs with different substituent position also showed different SOA formation characteristics. For xylene, the peak mass concentration of o-xylene (88.6 μg m$^{-3}$) was higher than that of m-

xylene and p-xylene, while the peak mass concentration of 123-TMB (82.0 μg m$^{-3}$) was significantly higher than those of



124-TMB (31.8 µg m$^{-3}$) and 135-TMB (27.6 µg m$^{-3}$) (Fig. 5b). These phenomena indicated that xylene and TMB with ortho methyl substituent facilitated the SOA formation. Further, the ortho methyl group of isomers (e.g. o-xylene, 123-TMB) could more thoroughly be oxidized, producing more particles (Sato et al., 2010).

It has also been reported that seed particles (e.g., NaCl) and highly relative humidity (up to 90%) can significantly
increase the yield of SOA (Wang et al., 2016;Luo et al., 2019;Jia and Xu, 2018). However, in this study, the maximum SOA yield of 25% (Fig. S11) was produced with increasing AH concentration, which was consistent with that from previous researches (Sato et al., 2012;Li et al., 2016;Song et al., 2007;Odum et al., 1997). Further considering the oxidation conditions of low RH (less than 5%) and seedless particles in this study, our results indicated that AH concentration should also be paid much attention to SOA formation although the addition of NO$_x$, seed particles and high RH are all very
important.

To further investigate the effect of AH's substituent on SOA yield, a two-product semi-empirical model was employed. As Fig. S12 shows, the model well fitted the SOA yield of nine AHs and the correspondingly fitting parameters are listed in Table 1. Similarly, high-volatile components were assumed from the photochemical oxidation of AHs and same $K_{om,2}$ of 0.005 m$^3$ µg$^{-1}$ was assigned. As seen from the table, benzene (0.242 m$^3$ µg$^{-1}$), toluene (0.162 m$^3$ µg$^{-1}$) and ethylbenzene
(0.422 m$^3$ µg$^{-1}$) showed higher $\alpha_2$ than that of xylenes (e.g., 0.086 m$^3$ µg$^{-1}$ for m-xylene) and TMBs (e.g., 0.082 m$^3$ µg$^{-1}$ for 123-TMB), indicating the production of more high-volatile products from AHs with less number of substituent. Meanwhile, benzene (0.022 m$^3$ µg$^{-1}$), toluene (0.027 m$^3$ µg$^{-1}$) and ethylbenzene (0.023 m$^3$ µg$^{-1}$) displayed lower $K_{om,1}$ in comparison with xylenes (e.g., 0.074 m$^3$ µg$^{-1}$ for p-xylene) and TMBs (e.g., 0.085 m$^3$ µg$^{-1}$ for 135-TMB), and the corresponding $\alpha_1$ decreased with increasing substituent number. All these results demonstrated that the increase of substituent number on phenyl ring
inhibited the generation of low-volatile products, thus reducing the generation of SOA particles, finally leading to the decrease of SOA yield. The result also indicated that the oxidation degree became lower and lower for AHs with increased substituent number, since the oxidation of methyl carbon was more difficult than that of carbon of phenyl ring. Li et al. reported similar phenomenon at early time,(Li et al., 2016) and is consistent with our results. However, they did not further investigate the relationship of isomer AH with the SOA yield.

In the present study, SOA yield of o-xylene was found higher than that of m-xylene and p-xylene, consistent with the SOA number and mass results. The fitting results showed that the $K_{om,1}$ of o-xylene (0.024 m$^3$ µg$^{-1}$) was much lower than that of m-xylene (0.057 m$^3$ µg$^{-1}$) and p-xylene (0.074 m$^3$ µg$^{-1}$), indicating the production of more low-volatile products from o-xylene. Similarly, 123-TMB showed the highest SOA yield and lowest $K_{om,1}$ among three TMBs. These results further confirmed that AHs with ortho methyl substituent favored the yield of SOA. This might be because that these AHs were
more susceptible to be oxidized, formed ring-opening products and finally produced more RO$_2$ than other isomers. Some previous studies have also obtained consistent results (Zhou et al., 2011;Song et al., 2007). In addition, ethylbenzene was also isomeric to xylene, and its SOA yield was higher than that of xylenes. Recent studies have found that the SOA yield during the oxidation of alkanes and alkenes by ˙OH increased with the carbon chain length (Lim and Ziemann, 2009;Tkacik et al., 2012). Obviously, the length of carbon chain also affected the oxidation degree of AHs. Then, the longer ethyl group



led to a higher degree of photochemical oxidation for ethylbenzene than xylenes, promoting the formation of more SOA precursors and finally higher SOA yield.

### 3.4. Enhanced formation mechanism of SOA with NOₓ

In order to further reveal the enhanced formation mechanism of SOA from the oxidation of AHs with the presence of NOx, the corresponding volatile intermediates were all identified, and quantified comparably. As Fig. 6a shows, with gradual

decrease of toluene concentration, the concentrations of small molecule carbonyl products, such as m31 (formaldehyde), m45 (acetaldehyde), m47 (formic acid) and m61 (acetic acid or glycolaldehyde), quickly increased to 16.0, 45.3, 31.0 and 17.0 ppb within 200 min. Acetaldehyde showed the highest concentration, which was by far higher than that obtained without $NO_x$ (9 ppb). Followed one were m85 (butanedione), m87 (butenedione) and m111 (hexene diketone) with the peak concentration below 2.8 ppb. The increase of concentration of m59 (glyoxal), m73 (methylglyoxal) and m99 (3-methyl-

2(5H)-furanone or 4-keto-2-pentenal) began to slow down after 120 min, and this trend was consistent with the trend of SOA formation (Fig. 5b). Similar variation trend of volatile intermediates for other AHs was also measured (Fig. S13). All these results demonstrated that there was a specific window period, and the intermediates in the gaseous phase were transformed into the particulate phase. The significant increase of SOA occurred after breaking through the window period.

Further comparison of volatile intermediates during photochemical oxidation of AHs with different VOC/NOₓ ratio was

also carried out. For toluene oxidation (Fig. 6b and 6c), the concentrations of all intermediates increased with the increase of VOC/NOₓ ratio. The carbonyl intermediates such as m59 (glyoxal) and m73 (methylglyoxal) were believed to be playing important role in the photochemical oxidation of AHs to form SOA (Li et al., 2016;Ji et al., 2017;Nishino et al., 2010). Bloss et al. also found glyoxal and methylglyoxal produced from toluene photochemical oxidation as the main precursor of SOA (Bloss et al., 2005b). In previous studies, the maximum yield of glyoxal and methylglyoxal were obtained as 20% and 17%

during toluene photochemical oxidation (Baltaretu et al., 2009;Volkamer et al., 2001;Nishino et al., 2010), which was lower than that of the present study (24% in Fig. 6c). Meanwhile, the yields of glyoxal and methylglyoxal during photochemical oxidation of toluene, xylenes and TMBs increased with increasing AH concentration. Therefore, the increase of AH content in reaction system promoted the photochemical oxidation of AHs to produce more volatile carbonyl intermediates, finally leading to higher SOA yield in this study. Moreover, the yield of m59 (glyoxal) and m73 (methylglyoxal) from benzene

photochemical oxidation was found the lowest among all AHs (Figs. 6b, 6c and S14-S22), indicating that the presence of branch chain on phenyl ring inhibited the production of unsaturated carbonyl compounds. This was because that increasing methyl group number of AHs weakened their oxidation reactivity, resulting in the inhibition of ring-opening reaction. Further, the formation of glyoxal and methylglyoxal from $RO_2$ were also subsequently suppressed. Furthermore, the yields of m85 (butanedione) from AHs containing ortho methyl group (e.g., o-xylene, 123-TMB and 124-TMB) were found higher

than that of their isomers, due to that ortho methyl groups of phenyl ring preferred to ring opening and then generation of ketone products (Li et al., 2016).





Based on the above results, the possible photochemical oxidation mechanism from AH to SOA were proposed. Toluene was selected as an example. As Fig. 7 shows, phenyl ring of toluene firstly reacted with ·OH to produce cresol (Ziemann, 2011;Atkinson, 2007), and then further oxidized to form an intermediate which could occur ring-opening reaction or react

with $HO_2$ radicals to form bicyclic peroxide compounds. The latter has been suggested as important SOA precursor from AH photochemical oxidation (Song et al., 2005;Wyche et al., 2009;Nakao et al., 2011;Johnson et al., 2005). The ring-opening intermediates were consisted of saturated and unsaturated dicarbonyl compounds (Jang and Kamens, 2001;Birdsall and Elrod, 2011). However, the possibility of these dicarbonyl intermediates directly partitioning into the particulate phase was very small (Jang and Kamens, 2001), but they could oligomerize to form low-volatility compounds (Forstner et al.,

1997;Jang and Kamens, 2001). The oligomerization was an important pathway for SOA formation from AH photochemical oxidation (Sato et al., 2012;Li et al., 2016;Hu et al., 2007). In our study, the detection of m85 (butanedione) and m99 (3-methyl-2(5H)-furanone or 4-keto-2-pentenal) proved the formation of ring-opening products. These unsaturated 1,4-dicarbonyls were observed to form small cyclic furanone compounds (Jang and Kamens, 2001;Bloss et al., 2005b). Therefore, the ring-opening products with saturated or unsaturated dicarbonyl groups finally transformed into SOA through

oligomerization process.

As mentioned above, with the increase of the substituent number on AHs, the yield of SOA decreased. The enhanced ring-opening products and restrained oligomerization reactions by the increased methyl group number were supposed to be the main cause. The methyl group is found to stabilize the ring-opening radicals (Ziemann, 2011). When phenyl ring contained methyl group, the oxidation pathway was prone to ring-opening. The concentrations of m87 and m111 increased

with the methyl group number increasing (Fig. S23), meaning that these two intermediates were dominant in the ring-opening products. However, they could not oligomerize to further partition into SOA formation (Fig. 7). Both non-cyclic dicarbonyls and cyclic compounds formed by unsaturated dicarbonyls were deemed to have small probability to oligomerize (Li et al., 2016;Kalberer et al., 2004). In previous study, no butanedione and hexene diketone were detected in particulate phase SOA under $300 \pm 1$ K and dry conditions (RH < 0.1%) in the absence of inorganic seed aerosol (Li et al., 2016).

However, butanedione was measured in gas phase of this study, indicating that it was not the precursor of SOA for the oligomerization reaction. The presence of methyl groups would inhibit the oligomerization to prevent the formation of ring compounds by unsaturated dicarbonyl groups and finally decrease SOA formation.

## 4 Conclusions

In this study, no SOA formation was observed from direct photochemical oxidation of AHs, while a small amount of $O_3$ was

produced without $NO_x$ addition. The presence of $NO_x$ significantly increased the productions of $O_3$ and SOA, due to synergetic effect of accelerated $NO_2$ conversion and AH reaction with NO as well as enhanced formation of volatile intermediates. Further increased formation of both $O_3$ and SOA were observed by promoted AH concentration. In addition, increase of AH's substituent number could enhance $O_3$ formation, but decrease SOA yield. The ortho methyl group



substituted AHs exhibited higher SOA yield. The preferential formation of variation of dicarbonyl intermediates and

restrained oligomerization reaction were responsible for above differences. These results showed more clear understanding of the effect of $NO_x$ and organic molecule structures on photochemical oxidation of AHs to form $O_3$ and SOA, which could provide solidly experimental basis for studying the transformation of AHs to secondary pollutants in the real atmospheric environment.

*Supplement*. The supplement related to this article is available online at:

*Author contributions*. TA designed research. HL and JC conducted experiments. HL, JC and GL analyzed data. HL prepared the manuscript with contributions from all co-authors.

*Competing interests*. The authors declare that they have no conflict of interest.

*Financial support*. This work was supported by the National Natural Science Foundation of China (41731279 and 42020104001), Local Innovative and Research Team Project of Guangdong Pearl River Talents Program (2017BT01Z032), and Guangdong Provincial Key

Research and Development Program (2019B110206002).

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





**Table 1.** Fitting yield curve parameters of two-product semi-empirical models

| AHs | $\alpha_1$ | $K_{om,1}$ $(m^3\ \mu g^{-1})$ | $\alpha_2$ | $K_{om,2}$ $(m^3\ \mu g^{-1})$ |
|---|---|---|---|---|
| Benzene | 0.341 | 0.022 | 0.242 | 0.005 |
| Toluene | 0.157 | 0.027 | 0.162 | 0.005 |
| Ethylbenzene | 0.285 | 0.023 | 0.422 | 0.005 |
| m-xylene | 0.103 | 0.057 | 0.086 | 0.005 |
| o-xylene | 0.345 | 0.024 | 0.017 | 0.005 |
| p-xylene | 0.085 | 0.074 | 0.057 | 0.005 |
| 123-TMB | 0.114 | 0.025 | 0.082 | 0.005 |
| 124-TMB | 0.068 | 0.075 | 0.080 | 0.005 |
| 135-TMB | 0.080 | 0.085 | 0.032 | 0.005 |






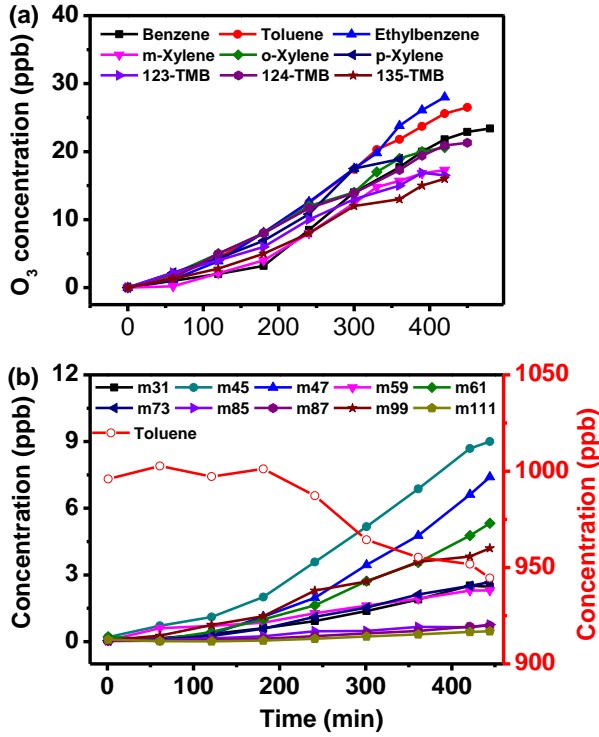

**Figure 1. (a) O₃ formation curve from AH photochemical oxidation (2110 ppb of benzene, 996 ppb of toluene, 1060 ppb of ethylbenzene, 889 ppb of m-xylene, 1160 ppb of o-xylene, 1040 ppb of p-xylene, 824 ppb of 123-TMB, 935 ppb of 124-TMB, 913 ppb of 135-TMB). (b) The concentration variation of intermediates formed from 996 ppb of toluene photochemical oxidation without NOₓ.**





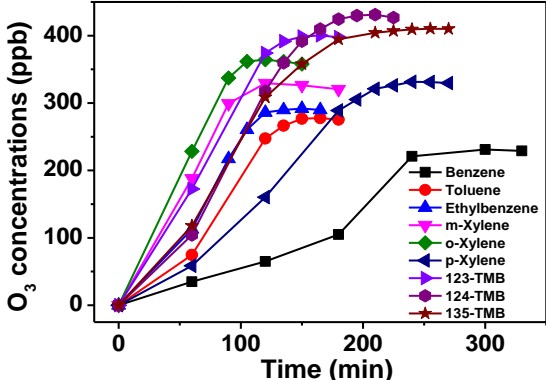

**Figure 2. O₃ formation curve from AH photochemical oxidation with the presence of NOₓ (2000 ppb of benzene and 160.2 ppb of NOₓ, 1048 ppb of toluene and 162.0 ppb of NOₓ, 1050 ppb of ethylbenzene and 162.4 ppb of NOₓ, 889 ppb of m-xylene and 172.1 ppb of NOₓ, 1052 ppb of o-xylene and 159.8 ppb of NOₓ, 1040 ppb of p-xylene and 157.2 ppb of NOₓ, 956 ppb of 123-TMB and 171.4 ppb of NOₓ, 1010 ppb of 124-TMB and 169.5 ppb of NOₓ, 1040 ppb of 135-TMB and 164.2 ppb of NOₓ).**



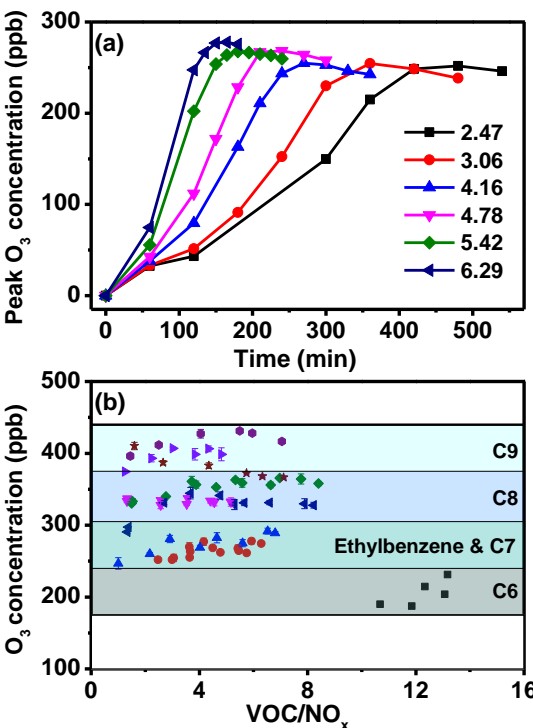

**Figure 3. (a) O₃ formation curve of toluene photochemical oxidation at different VOC/NOₓ ratio and (b) change trend of the peak O₃ generated by photochemical oxidation of AHs at different VOC/NOₓ ratio (square: benzene; circle: toluene; upper triangle: ethylbenzene; lower triangle: m-xylene; diamond: o-xylene; left triangle: p-xylene; right triangle: 123-TMB; hexagon: 124-TMB; pentacle: 135-TMB).**




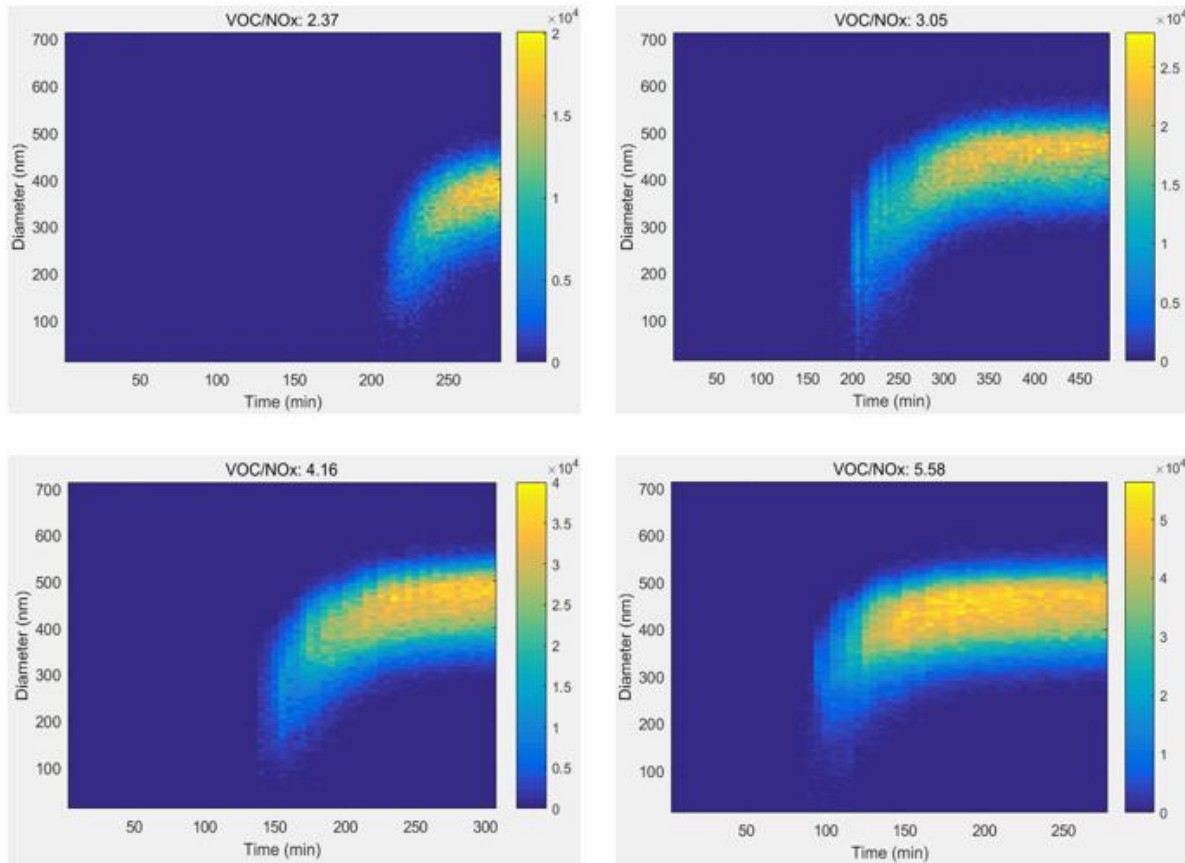

**Figure 4. Nanoparticle distribution from toluene photochemical oxidation varied with time at different VOC/NO$_x$ ratio (the initial concentrations of NO$_x$ were in the range of 158.2 to 179.7 ppb).**





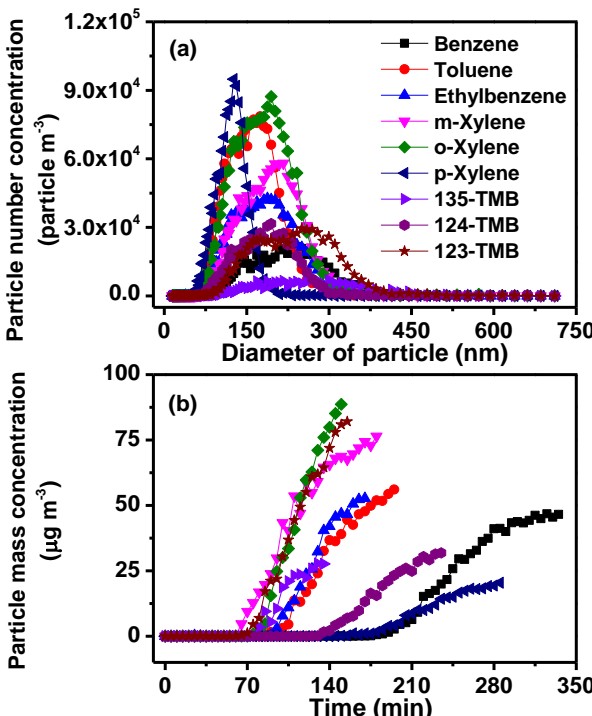

**Figure 5. (a) Number concentration and (b) mass concentration of SOA produced from AH photochemical oxidation with the presence of NO$_x$ (2000 ppb of benzene and 160.2 ppb of NO$_x$, 1048 ppb of toluene and 162.0 ppb of NO$_x$, 1050 ppb of ethylbenzene and 162.4 ppb of NO$_x$, 889 ppb of m-xylene and 172.1 ppb of NO$_x$, 1052 ppb of o-xylene and 159.8 ppb of NO$_x$, 1040 ppb of p-xylene and 157.2 ppb of NO$_x$, 956 ppb of 123-TMB and 171.4 ppb of NO$_x$, 1010 ppb of 124-TMB and 169.5 ppb of NO$_x$, 1040 ppb of 135-TMB and 164.2 ppb of NO$_x$).**





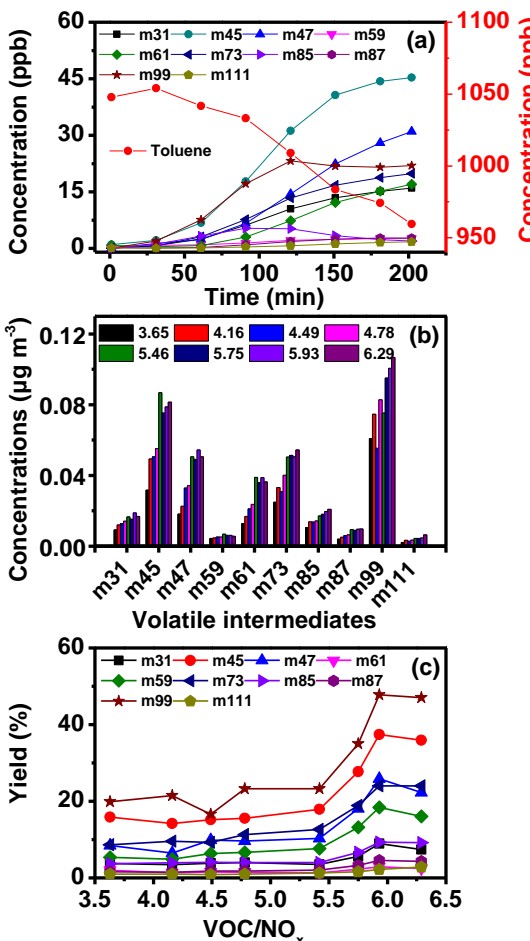

**Figure 6.** Intermediates formed from toluene photochemical oxidation in the presence of NO$_x$ (a) concentration variation with the reaction time (1048 ppb of toluene and 162 ppb of NO$_x$) (b) concentration and (c) yield with the increase of VOC/NO$_x$ ratio.






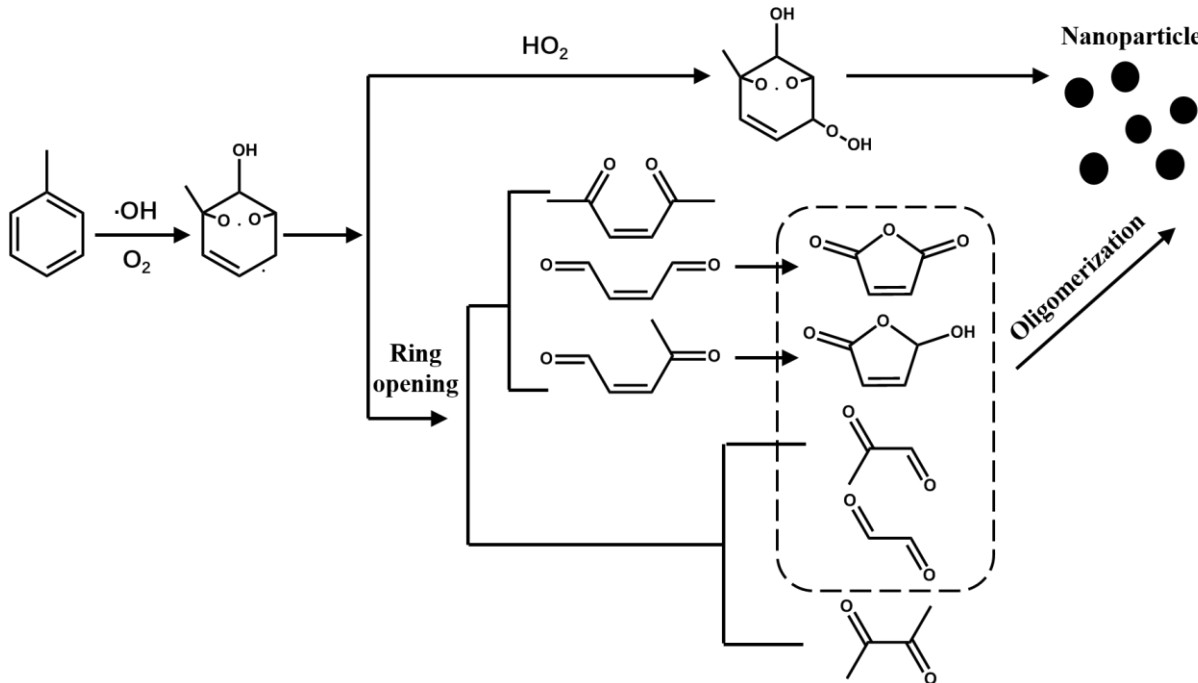

**Figure 7. The possible photochemical oxidation mechanism of toluene to SOA formation.**