# Peer review of "Supporting information"

_Atmospheric Chemistry and Physics, 2021_

## Author Response (AR2)

**Manuscript ID: acp-2021-29**

**Title:** Formation kinetics and mechanism of ozone and secondary organic aerosols from photochemical oxidation of different aromatic hydrocarbons: dependence on NOx and organic substituent

**The corresponding author:** Prof. Taicheng An

**Dear editor, Anonymous Referees, and Prof. Liming Wang,**

We sincerely thank you for the critical comments and thoughtful suggestions from you and the reviewers that we have used to improve the quality of our manuscript. We have carefully revised the manuscript in accordance with your and reviewers' comments. We are now submitting the revised version of the manuscript. The responses to reviewers' comments are attached. A marked copy highlighting the changes made in the text is enclosed separately. We hope the revised paper will meet your standard. Below you will find our point-by-point responses to the reviewer's comments:

**Anonymous Referee #1**

*Question 1: The title may be changed to '... dependence on Nx and organic substituent'.*

**Response:** We are very grateful to the reviewer's suggestion. The title is accordingly revised.

**Author's changes in manuscript:** The new title was 'Formation kinetics and mechanism of ozone and secondary organic aerosols from photochemical oxidation of different aromatic hydrocarbons: dependence on NOx and organic substituent' in the revised manuscript. Please see line 1-3 in the revised main manuscript and supporting information.

*Question 2: Line 70: NO can photolyze under UV irradiation with a wavelength of less than 420 nm. Please indicate the centre of the UV lamp wavelength applied in this work in the experimental section.*

**Response:** We are very grateful to the reviewer's comment. The centre of the UV lamp wavelength applied in this work was 360 nm, which was accordingly supplied in

the experimental section of the revised manuscript.

**Author's changes in manuscript:** The sentence 'The centre of the UV lamp wavelength was 360 nm.' was provided in the revised manuscript. Please see line 72 in the revised main manuscript.

*Question 3: Line 80, 88-90: The detection limit of instruments should be provided. Line 123: Did the authors detect any oligomer products during the photochemical reaction in this work?*

**Response:** We are very grateful to the reviewer's comment. The detection limits of PTR-ToF-MS (< 20 ppt for m/z 79 and < 10 ppt for m/z 181 within averaged over 1 min), NOx analyzer (< 0.4 ppb within averaged over 1 min) and $O_3$ analyzer (0.5 ppb) were accordingly provided in the revised manuscript. And a total of ten products were in-situ detected by PTR-ToF-MS in gas phase (shown in Figure S14). However, no oligomer products were detected during the photochemical reaction in this work, probably due to their low concentration or easily being adsorbed by the reactor.

**Author's changes in manuscript:** The sentence 'The detection limit of PTR-ToF-MS was < 20 ppt for m/z 79 and < 10 ppt for m/z 181 within averaged over 1 min).' was provided in the revised manuscript. Please see line 87-88 in the revised main manuscript.

*Question 4: Line 160-165: As the authors suggested that reaction conditions, such as the VOC/NOx ratio, could influence the formation rate and mechanism of O3. What are the conditions of the experiments in previous studies?*

**Response:** We are very grateful to the reviewer's comment. The VOC/NOx ratio ranging from 1.0 to 13.0 was selected to evaluate its effect to $O_3$ formation. Thus, the corresponding range of VOC/NOx ratio in our work and main references were collected and listed in Table S2 of revised supporting information. After comparison, it was found that the range of VOC/$NO_x$ ratio in these references were close to that in our study.

**Author's changes in manuscript:** The sentence in Line 164-165 of original manuscript was revised as 'Meanwhile, the VOC/NOx ratio ranging from 1.0 to 13.0 was selected to its effect to $O_3$ formation. And the range of VOC/$NO_x$ ratio in above researches was close to that in our study (Table S2). Then, our results of $O_3$ concentration were comparable to those in the previous studies under similar range of VOC/$NO_x$ ratio'. Please see line 166-169 in the revised main manuscript.

**Question 5:** *Line 497: Please clarify the reaction time in Figure 5 (a).*

**Response:** We are very grateful to the reviewer's comment. The number concentrations of SOA in Figure 5(a) were obtained at the endpoint of each reaction. Accordingly, the clarification was provided.

**Author's changes in manuscript:** 'The particle number concentration of SOA was obtained at the endpoint of each reaction.'. Please see line 188-189 in the revised main manuscript.

**Question 6:** *Some spelling mistakes should be avoided in the revised version, such as "NO2" in Line 18.*

**Response:** We are very grateful to the reviewer's comment. The whole manuscript was carefully checked and the mistakes were accordingly revised.

**Anonymous Referee #2**

**Question 1:** *The paper requires some general proofreading for English grammar.*

**Response:** We are very grateful to the reviewer's comment. The whole manuscript was carefully checked and the spelling mistakes were accordingly revised.

**Question 2:** *Please add a Table containing initial experimental conditions in Supporting Information in case that the reader can be repeated the procedure as well as the results.*

**Response:** We are very grateful to the reviewer's suggestion. Typical experimental conditions of this study for nine AHs were supplied in the revised supporting information as listed in Table S1.

**Author's changes in manuscript:** 'Typical experimental conditions (e.g., concentrations of AHs and NOx, VOC/NOx ratio, RH and temperature) of this study for nine AHs were supplied in Supporting Information (SI) as Table S1'. Please see line 77-79 in the revised main manuscript.

*Question 3: Line 146: The change of VOC/NOx ratio is the key factor affecting the transformation of AHs to O3. The similarity and difference between this study and previous others' work should be described*

**Response:** We are very grateful to the reviewer's comment. The VOC/NOx ratio ranging from 1.0 to 13.0 was selected to evaluate its effect to $O_3$ formation. Thus, the corresponding range of VOC/NOx ratio in our work and main references were collected and listed in Table S2 of revised supporting information. The accordingly comparison of our results with these previous data were then conduced in the revised manuscript.

**Author's changes in manuscript:** 'Meanwhile, the VOC/NOx ratio ranging from 1.0 to 13.0 was selected to its effect to $O_3$ formation. And the range of VOC/$NO_x$ ratio in above researches was close to that in our study (Table S2). Then, our results of $O_3$ concentration were comparable to those in the previous studies under similar range of VOC/$NO_x$ ratio'. Please see line 167-169 in the revised main manuscript.

*Question 4: Abstract: Change "NO2" to "$NO_2$".*

**Response:** We are very grateful to the reviewer's suggestion and the corresponding correction was made. Please see line 18 in the revised main manuscript.

*Question 5: Line 48 and 50: "•OH-initiated" should be replaced with "OH-initiated".*

**Response:** We are very grateful to the reviewer's suggestion and the correction was

accordingly made. Please see 48 and 50 in the revised main manuscript.

**Question 6:** *Line 100: Change "0 ppb to 16" to "0 to 16 ppb".*

**Response:** We are very grateful to the reviewer's suggestion and corresponding modification has been made. Please see line 103 in the revised main manuscript.

**Question 7:** *Line 132: The reason of choosing 160 ppb NO2 added in the chamber should be given in the manuscript.*

**Response:** We are very grateful to the reviewer's comment. The concentration of $NO_2$ was selected based on previous works. In these two works, about 100 - 200 ppb of NOx was applied to investigate the photochemical oxidation of AHs. Therefore, the middle value of this region (e.g., $160 \pm 10$ ppb) was accordingly chosen in this study. Accordingly, the reason was provided in the revised manuscript.

**Author's changes in manuscript:** 'The concentration of NO2 was selected based on previous works (Luo et al., 2019; Chen et al., 2018)'. Please see line 36 in the revised main manuscript.

**Prof. Liming Wang's comments:**

**Question 1:** *The PTR measurements provided some information on the reaction intermediates, but the PTR-MS data were hardly discussed in this manuscript. Simple description on concentration changes for products and peak concentrations of $O_3$ is NOT directly relevant to $O_3$ or SOA formation. No discussion on the formation mechanism of acetaldehyde, formic acid, and acetic acid which were not usually observed in previous studies. Effort might be required to work out how these compounds are formed by referring to previous publications or suggesting new formation pathways.*

**Response:** We are very grateful to the reviewer's comment. As known, AHs contribute significantly to $O_3$ and SOA formation in atmosphere, but the effect of AH's

substituent to their formation are still not clear well. In this study, our results revealed that $O_3$ formation was enhanced with increasing AH's substituent number but negligibly affected by their substituent position. Differently, SOA yield decreased with the increased substituent number of AHs, but increased with ortho methyl group substituted AHs. In order to establish the relationship between AH oxidation and $O_3$ and SOA formation, the oxidation products of AHs with different structures were on-line detected by using PTR-TOF-MS. By combining intermediate evolution results as well as model fitting data, it consistently confirmed that increasing substituent number on phenyl ring inhibited generating dicarbonyl intermediates, which however were preferentially produced from oxidation of ortho methyl group substituted AHs, resulting in different changing trend of $O_3$ and SOA yield. Actually, in this study, the products from PTR-TOF-MS were both deduced from their m/z information and also compared with those from previous publications. And thanks to the reviewer suggestion, our recent work is conducting on the in-situ formation kinetics and mechanism of intermediates by using a combined fast flow-tube with Chemical ionization mass spectrometer. The corresponding results will be published in our next paper.

***Question 2:*** *The chamber studies here used rather high concentrations of AHs (~1000 ppbv or higher) and initial NOx (~100 ppbv or higher). Unfortunately, NOx concentrations were not reported in the course of reactions. Both [AH] and [NOx] are considerably higher than values in usual atmosphere.*

**Response:** We are very grateful to the reviewer's comment. In this study, we tried to establish the relationship between AHs and $O_3$ and SOA formation. Under the concentration region of our selected, it was found that the structure of AHs indeed showed different contributions to $O_3$ and SOA formation. As the reviewer mentioned, the atmospheric concentrations of AHs and $NO_X$ were actually at ppb- or ppt-level, while the detection limits of equipment used in this study were also at similar level. In addition, in our preliminary experiments, the reaction of

same leveled AHs and NOx were investigated. However, the obtained results were lack of reliability, due to big variation of detected data. Therefore, relatively higher concentrations of AHs and NOx were chosen to establish the relationship between AHs and $O_3$ and SOA formation. Meanwhile, the concentrations of AHs and $NO_x$ were selected based on the previous works [1, 2]. In these two works, about 100 - 200 ppb of NOx was applied to investigate the photochemical oxidation of AHs (500 - 2000 ppb). Therefore, the middle values of this region (e.g., 160 ± 10 ppb and 1000 ppb) were accordingly chosen in this study.

***Question 3:*** *Product identification by PTR-MS: (a) Line 118/233/254: m/z 85 should be "2-butenedial", m/z 87 might be "butanedione". (b) Line 230: Why was m/z 111 assigned to "hexene diketone"? I am not sure what "hexane diketone" means. I suppose it be CH3C(O)CH=CHC(O)CH3, and its cation C6H9O2 + after PTR has m/z 113.*

**Response:** We are very grateful to the reviewer's comments. We have carefully checked the original data and found that the name of products m/z 85 and 87 were miswritten. We corrected them in the revised manuscript: m/z 85 (2-butenedial) and m/z 87 (butanedione). For m/z 111, hexa-2,4-dienedial was temporarily assigned as ring-opening product of AHs after carefully analyzing the mass information and comparing with some previous works. All these modifications were marked as yellow color in the revised version. Deeper investigation was conducting in our recent study to further confirm our current results and the corresponding results will be published in our next paper.

***Question 4:*** *We notice the experiments were carried out in a 2 m3 chamber. Was the wall effect corrected? Aerosol and $O_3$ can also lose on the reactor wall on long reaction time.*

**Response:** We are very grateful to the reviewer's comment. The reactor characterization and primary application of dual-reactor chamber in the investigation of atmospheric photochemical processes has been investigated and the corresponding results

were published in the Journal of Environmental Sciences (2020, 98: 161-168). The preliminary work also revealed that the chamber could be well utilized to simulate gas-particle conversion progresses in the atmosphere. In this study, the wall loss of all AHs, their products, $O_3$ and SOA were all characterized by using same methods in this published paper and the data shown in the manuscript were corrected with wall loss.

***Question 5:*** *How was OH radical generated? Particularly for NOx-free experiments. This should be important in understanding $O_3$ formation under NOx-free conditions.*

**Response:** We are very grateful to the reviewer's comment. Actually, in order to avoid the effect of OH radical in our experiments, we deliberately controlled the experimental relative humidity lower than 5%. Although OH radical may be generated from AH photolysis, it was believed that its production was very low to trigger reaction to form intermediates and then $O_3$.

In this study, the possible contributors of these $O_3$ were intermediates such as carbonyl compounds from AH photolysis when the absent of NOx. In all, our results indicated that direct photochemical transformation of AHs to $O_3$ actually occurred and should be taken into consideration in the atmospheric environment. More works should be done to comprehensively reveal the formation mechanism of $O_3$ from AH photolysis.

***Question 6:*** *Figures: All $O_3$ formation curves are presented as [$O_3$] vs Time. Different aromatic benzenes have different reactivity towards OH radical. Besides, the initial [AH] concentrations are different. Therefore plotting "[$O_3$] vs Time" does not provide too much insight on the progress of the reaction. It is probably more proper to present "[$O_3$] vs D[AH]", which might give information on $O_3$ formation potential. Similarly for concentrations of other products. Besides, on shortening of $O_3$ peak appearance when increasing AH concentrations (Line 146-154, Figure 3a) might also be rationalized in terms of AH consumption.*

**Response:** We are very grateful to the reviewer's suggestion. In this work, we tried to reveal

the formation kinetics of $O_3$ from photochemical oxidation of different AHs. To more easily understand the formation trend of $O_3$ and also compare the formation concentration of $O_3$ at different reaction time for different AHs, all $O_3$ formation curves were presented as $O_3$ concentration vs reaction time. Similarly, this expression was also applied for the concentrations of products. This was also due to the very close initial concentration of almost all AHs (except for benzene), making them comparable. In Figure 3a, by using same plotting method, we clearly showed reader the different time for reaching the maximum $O_3$ concentration toward different AHs.

**Question 7:** *Section 3.1 The First Paragraph: The rate coefficients of AHs with OH radical were well known. No need to confirm their reactivity.*

**Response:** We are very grateful to the reviewer's comment. The rate coefficients of AHs with OH radical was cited from the previous works, and used to support the reactivity of AHs with different structure. Moreover, the corresponding data shown here could make the reader quicker understand the order of reactivity for different AHs.

**Question 8:** *Section 3.1 "… without NOx": I am surprised to see $O_3$ formation under NOx-free conditions (Figure 1a). The $O_3$ concentration is quite substantial, up to 25 ppbv with a consumption of ~50 ppbv in toluene. Quite high yields here. Our current understanding of $O_3$ formation is based on photolysis of $NO_2$. The authors stated "The possible contributors of these $O_3$ might be intermediates such as carbonyl compounds" but offered no details. Could the authors be more specific and give possible reactions leading to ozone formation? This is really important if the measurements are correct here!*

**Response:** We are very grateful to the reviewer's comment. Normally, the formation of $O_3$ was from photolysis of $NO_2$ according to equations shown in section 3.2. However, in our work, the gradually increased concentration of $O_3$ was truly found under the direct photochemical oxidation of ALL nine AHs. Meanwhile,

the formation of ten products was detected. Moreover, the increased trend of $O_3$ and some intermediates (such as m45, m47, m61, m99) of AHs was consistent. And this reaction system was lack of $NO_2$. Therefore, we proposed that "the possible contributors of these $O_3$ might be intermediates such as carbonyl compounds". Since we noticed this surprised results, the further investigation was conducting to reveal the relationship of these intermediates and $O_3$ formation.

***Question 9:*** *Line 139: "... AHs ... reduce the consumption of $O_3$". This is NOT correct. In the oxidation of VOCs, the increased $O_3$ formation is due to the addition conversion of NO to $NO_2$ by $RO_2$/$HO_2$ radicals.*

**Response:** We are very grateful to the reviewer's comment. In general, photolysis of $NO_2$ led to the formation of $O_3$ via equations 1-3 as shown in section 3.2 in this manuscript. When the presence of AHs, $O_3$ would degrade AHs to form $RO_2$ and $HO_2$, both of which then competed with $O_3$ to react with NO, leading to the reduced consumption of $O_3$. Hence, we concluded "the presence of AHs could compete with $O_3$ for the NO reaction and reduce the consumption of $O_3$.".

***Question 10:*** *Figure 1: Specify "NO free" in figure caption. Figure 2: It might be necessary to plot the concentrations of AH on the same plot.*

**Response:** We are very grateful to the reviewer's comment. We added "without $NO_x$" in figure caption of Figure 1a. The variations of concentration of AHs were plotted in Figures 1b and S1, so we did not repeatly provided them in Figure 2.

***Question 11:*** *Line 155: I am not sure the purpose to compare $O_3$ peak concentrations. I believe what we really care is the yield, instead of peak concentrations under some particular reaction conditions (VOC concentration, or VOC/NOx ratios, ...).*

**Response:** We are very grateful to the reviewer's comment. In this work, we tried to reveal the formation kinetics of $O_3$ from photochemical oxidation of different AHs. To achieve this, the $O_3$ peak concentrations of nine AHs obtained at the same

VOC/NO$_x$ ratio were compared as an example. We also compared our results with previous one and found that previous studies only focused on one or several AHs, and the relationship between AH substituent and O$_3$ formation was still not fully understood. Our results obtained in this study clearly confirmed that increasing substituent number of AH correspondingly increased O$_3$ concentration. It was also noticed that the O$_3$ peak concentrations of xylene or TMB isomers were in the same range, suggesting negligible effect of substituent position of AHs to their O$_3$ formation. Therefore, discussing O$_3$ peak concentration obtained from different AHs could efficiently show readers relationship between different AHs and O$_3$ formation.

***Question 12:*** *Line 250-251: Yields of Gly and MGly in benzene are lower than those in substituted benzenes. Then what does "inhibited" mean?*

**Response:** We are very grateful to the reviewer's comment. In order to avoid ambiguity, we revised the word "inhibited" to "did not favour" in the revised manuscript and marked as yellow color. Please see line 254 in the revised version.

***Question 13:*** *Line 253: "... resulting in the inhibition of ring-opening reaction". Any evidence for this claim? As far as I know, ring-opening of the bicyclic alkoxy radical is faster when alkyl-substitution is available.*

**Response:** We are very grateful to the reviewer's comment. Li et al. has reported that due to the increase of aromatic groups, the oxidation of AHs was weakened, resulting in the formation of incomplete oxidation products and the inhibition of ring opening reaction, which has been confirmed by (H/C and O/C) elemental analysis in previous study [3]. We then cited this reference in the revised manuscript to support our hypothesis.

***Question 14:*** *Line 263-165: "However, ... Kamens, 2001)." The sentence gives an impression that oligomerization occurs before partitioning into particles. The fact is that dicarbonyls are uptaken by particles first and then dimerize in particles.*

**Response:** We are very grateful to the reviewer's suggestion. In this study, we mentioned that the possibility of these dicarbonyl intermediates directly partitioning into the particulate phase was very small. This means that most of intermediates were in gas phase. It does not mean that oligomerization occurred before partitioning into particles. Actually, at the end of this paragraph, we concluded that "Therefore, the ring-opening products with saturated or unsaturated dicarbonyl groups finally transformed into SOA through oligomerization process.". Please see line 272-273 in the revised version.

***Question 15:*** *Line 274-276: Formation of m/z 87 and m/z 111 depends more strongly on the pattern of methyl substitution, but only on the number of methyl substitution. m/z 87 from neighboring methyl substitutions (o-Xylene and 1,2,3-TMB), while m/z 111 from meta-methyl pairs (m-Xylene and 1,3,5-TMB). Again Fig 23S was plotted as "concentration vs time". High concentration does NOT necessarily means high yield, and discussions on them unlikely lead to reliable conclusion.*

**Response:** We are very grateful to the reviewer's comment. In this paragraph, we tried to establish the relationship between SOA formation and important ring-opening intermediates. It was found that the enhanced ring-opening products and restrained oligomerization reactions by the increased methyl group number were supposed to be the main cause for SOA formation. The methyl group was found to stabilize the ring-opening radicals. When phenyl ring contained methyl group, the oxidation pathway was prone to ring-opening. The concentrations of m87 and m111 increased with the methyl group number increasing, meaning that these two intermediates were dominant in the ring-opening products. However, they could not oligomerize to further partition into SOA formation. And our conclusion was that the presence of methyl groups would inhibit the oligomerization to prevent the formation of ring compounds by unsaturated dicarbonyl groups and finally decrease SOA formation.

**References**

[1] H. Luo, L. Jia, Q. Wan, T. An, Y. Wang, Role of liquid water in the formation of O3 and SOA particles from 1,2,3-trimethylbenzene, Atmospheric Environment, 217 (2019) 116955.

[2] Y. Chen, S.R. Tong, J. Wang, C. Peng, M.F. Ge, X.F. Xie, J. Sun, Effect of Titanium Dioxide on Secondary Organic Aerosol Formation, Environ Sci Technol, 52 (2018) 11612-11620.

[3] L. Li, P. Tang, S. Nakao, C.L. Chen, D.R. Cocker, Role of methyl group number on SOA formation from monocyclic aromatic hydrocarbons photooxidation under low-NOx conditions, Atmos Chem Phys, 16 (2016) 2255-2272.